# Effects of Population Regulation on the Source–Sink System of Hybrid Wheat Jingmai 6

**Weibing Yang** **, Zheng Wang, Liping Ren, Zhijie Ye, Xinhuan Gao, Jiangang Gao, Hongyao Lou, Bing Du** , **Zhaobo Chen \* and Shengquan Zhang \***

Institute of Hybrid Wheat, Beijing Academy of Agricultural and Forestry Sciences, Beijing 100097, China
\* Correspondence: chzhaobo@126.com (Z.C.); zsq8200@126.com (S.Z.)

**Abstract:** Hybrid wheat is considered to be one of the main ways to greatly improve the wheat yield in the future, and population construction is an important factor affecting their yield heterosis formation. In order to clarify the effect of population regulation of hybrid wheat on source–sink systems, the two-line hybrid wheat variety Jingmai 6 was used to carry out relevant research in this experiment. The leaf area index (LAI) of Jingmai 6 showed an increasing trend, while the tiller-spike rate exhibited a downward trend with the increase of planting density. The total accumulation of dry matter at maturity increased with the increase of planting density, among which the relative proportion of dry matter pre-anthesis gradually increased, while that after anthesis gradually decreased. The sink capacity and spikes number of hybrid wheat were positively correlated with the grain yield. There was higher sink capacity, spike to leaf ratio and grain to leaf ratio under A2 and A3 conditions. With the increase of planting density, the hybrid wheat yield showed a unimodal curve, and A3 had the highest grain yield. Combined with the change trend of dry matter accumulation before and after anthesis, it was proved that suitable planting density was the key to obtaining high yield of hybrid wheat.

**Keywords:** hybrid wheat; population regulation; source–sink relations; yield

## 1. Introduction

One of the most significant applications of hybridization breeding was the utilization of heterosis, and the great success that has been achieved in maize and rice [1–3]. However, hybrid wheat has not been very successfully used in large-scale wheat production worldwide. At present, success and popularity of hybrid wheat has been witnessed in parts of Western Europe, India, China and USA [4]. As reported, the area of hybrid wheat in Europe increased from 100,000 ha in 2002 to 560,000 ha in the year 2017–2018 [4], and hybrid varieties occupied >60,000 acres in six states by 2004 in India. The photo-thermo-sensitive male sterile lines, with the features of full sterility in hybrid seed production area and fully fertile hybrids in farmers' fields, have been discovered by our team and the technical system of two-line hybrids wheat in China was first established in 2011, which offered prospects for the production of hybrid wheat on a commercial scale. In the past decade, China's two-line hybrid wheat has made important progress in breeding. More than 10 hybrid wheat varieties have passed the national variety approval and their plant area is gradually increasing. Among them, the yield of Jingmai 6 was more than 10% higher than that of the control, and also showed strong characteristics of salt tolerance [5]. A mid-parent heterosis for grain yield of approximately 10% was also reported for hybrid bread and durum wheat [6,7]. In addition, because of the yield increase of 3.5–15% and other advantages, including abiotic and biotic stresses tolerance, the promising heterosis is still considered to be one of the main ways to greatly improve the wheat yield [5], and may make important contributions for feeding a growing population [8,9]. Grain yield of wheat is considered as the balance between the photo-assimilate supply (source) and the capacity of the grains to accumulate the photo-assimilates (sink) [10], and the source–sink relations is of great significance to

explain the formation of crop yield [11–13]. Source–sink relations can be affected by many factors, including the plant type, climate conditions and fertilization management [14], and the reasonable cultivation is helpful to optimize the source–sink relations, and then to improve the grain yield [15,16]. The results showed that the tiller–spike rate of hybrid wheat was high. Reasonable reduction of density and full utilization of higher tiller–spike rates were beneficial to shaping a reasonable population [17–19]. Some studies also found that hybrid wheat has a large leaf source and sufficient supply of assimilates, which can lay a foundation for grain filling [20,21]. The potential advantage of sink capacity is better than that of conventional wheat [22], which helps to increase the mobilization of source materials and improve the contribution of pre-anthesis storage materials of hybrid wheat to grain formation under adverse conditions [23–26] It also shows obvious yield increase advantage under medium- and low-yield conditions. At present, the leaves of hybrid wheat show the characteristics of large, wide and scattered. The larger population is easy to cause leaf self-shading within the canopy, which is not conducive to the production of photosynthetic materials and then affects the grain filling. It is of great significance to construct a reasonable population for hybrid wheat in the field production. However, the relations of population construction to source–sink relations and grain yield of hybrid wheat under field production conditions is still unclear.

In this study, Jingmai 6 was used as the experimental material. Under the condition of high yield, the planting density recommended was 1.50–2.25 million basic seedlings ha$^{-1}$. Based on the recommended planting density, we set up different density experiments to systematically reveal the yield formation mechanism from yield composition, source–sink relations, material accumulation and distribution, etc. Moreover, these can also provide technical support for the demonstration and promotion of the same type of hybrid wheat varieties in the future.

## 2. Materials and Methods

### 2.1. Plant Materials and Experimental Design

The experiments were carried out at the Experimental Station of Institute of Hybrid wheat in Shunyi, Beijing, China (40°13′ N, 116°65′ E). From 5 October 2015 to 28 June 2016, the hybrid wheat of Jingmai 6 (BS210 × F832) were used to carry out the analysis of source–sink relations and grain yield under different plant density. Maize (*Zea mays* L.) was the previous crop, and the soil type is sandy loam. In the growing season, 600 kg ha$^{-1}$ of special compound fertilizer for wheat (N:P:K, 16:21:8) was applied as basal fertilizer before planting, with 150 kg N ha$^{-1}$ being top-dressed at the jointing stage [27]. There were no noticeable crop damages from weeds, insects or diseases, and no special weather events during the grain filling stage.

Six plant densities were designed in this experiment, which were A1 (75 × 10$^4$ basic seedlings ha$^{-1}$), A2 (150 × 10$^4$ basic seedlings ha$^{-1}$), A3 (225 × 10$^4$ basic seedlings ha$^{-1}$), A4 (300 × 10$^4$ basic seedlings ha$^{-1}$), A5 (375 × 10$^4$ basic seedlings ha$^{-1}$) and A6 (450 × 10$^4$ basic seedlings ha$^{-1}$). Each treatment was repeated four times. In order to ensure the accuracy of the grain yield and other data, the method of small plot planting was adopted, with an area of 7.2 m$^2$ (9 rows, the planting row spacing is 16cm, and the length is 5 m).

### 2.2. Measurements and Analyses

#### 2.2.1. Population Stems and Tillers Investigation

The basic seedlings shall be investigated at the three-leaf stage of wheat, and plots with uniform growth shall be selected in each plot (the investigation area of each plot is 0.32 m$^2$). The changes of tillers were investigated at the main growth stages of wheat, and the tiller–spike rate was calculated [28]. The calculation formula is as follows:

- Tiller–spike rate (%) = (Spike number at maturity − Basic Seedling)/(maximum total stem number − Basic Seedling) × 100

### 2.2.2. Canopy Characters

Thirty plants were randomly obtained from each plot during the anthesis stage. The length (L, cm) and maximum width (W, cm) of each leaf was determined with a ruler. Leaf area was calculated as follows: Leaf area (cm$^2$) = 0.83 × L × W. The LAI was calculated as follows: LAI = leaf area × N/S, where N is the number of wheat plants per unit area of field and S is the unit area of field [29]. The spike-to-leaf ratio and grain-to-leaf ratio were also calculated by the following equation [30]:

- The spike-to-leaf ratio = The number of spikes per unit land area/the sum of top three leaf area per unit land area during the anthesis stage
- The grain-to-leaf ratio = Number of grains per unit land area/the sum of top three leaf area per unit land area during the anthesis stage

### 2.2.3. Material Accumulation Dynamics

In the main growth period, the typical plants (0.32 m$^2$) were taken back to the room and divided into three parts: leaf, stem sheath and spikes. It was killed at 105 °C for 30 min and dried at 80 °C to constant weight. The dynamics of dry matter accumulation of the population were investigated and the relevant indexes were calculated [31,32]. The calculation formula is as follows:

- Relative proportion of dry matter after anthesis (%) = (Dry matter weight of population at mature − Dry matter weight of population at anthesis)/Dry matter weight of population at mature × 100
- Relative proportion of dry matter pre-anthesis (%) = (Dry matter weight of vegetative organs at anthesis − Dry matter weight of vegetative organs at maturity)/Dry matter weight of vegetative organs at anthesis × 100
- Contribution proportion pre-anthesis (%) = (Dry matter weight of vegetative organs at anthesis − Dry matter weight of vegetative organs at mature)/Grain weight × 100
- Dry matter productivity after anthesis (g/m$^2$) = Dry matter accumulation after anthesis/Total leaf area at anthesis
- Dry matter mass per unit grain before anthesis (mg/grain) = Dry matter weight at anthesis/Total number of grains
- Dry matter mass per unit grain after anthesis (mg/grain) = (Dry matter weight at maturity − Dry matter weight at anthesis)/Total number of grains.

### 2.2.4. Yield Determination

Harvest each plot at the mature period, with a harvest area of 6 m$^2$. After harvest, dry and weigh, and then measure the water content, which is converted into the yield under the condition of 13% water content. The population pool capacity is calculated as follows [33]:

Storage capacity (×10$^7$ grains ha$^{-1}$) = Number of spike per unit land area × Number of seed set per spike.

### *2.3. Statistical Analysis*

The analysis of variance was also performed by Data Processing System (DPS) software (Version 7.05, Hangzhou, China). Data from each sampling date were analyzed separately. Means were tested by the least significant difference at the *p* < 0.05 level (LSD 0.05).

## 3. Results
### *3.1. Characteristics of Population Structure of Hybrid Wheat*
### 3.1.1. Population Leaf Area Index (LAI) at Anthesis

LAI is a good dynamic index to reflect the size of crop population. With the increase of sowing density, the LAI of hybrid wheat at anthesis stage showed an increasing trend (Table 1). A5 and A6 had the highest LAI of total population (4.75), while that of A1 was the lowest (3.53). Similarly, the LAI of the top 3 leaves showed the same trend.

**Table 1.** Composition of LAI at anthesis in different densities.

| Treatments | Basic Seedling ($\times 10^4$ ha$^{-1}$) | Total Population | Top 3 Leaves |
|---|---|---|---|
| A1 | 75.0 | 3.53 ± 0.18 d | 2.75 ± 0.20 b |
| A2 | 150.0 | 3.84 ± 0.07 c | 2.71 ± 0.16 b |
| A3 | 225.0 | 4.06 ± 0.13 b | 2.79 ± 0.23 b |
| A4 | 300.0 | 4.27 ± 0.09 b | 3.35 ± 0.17 a |
| A5 | 375.0 | 4.75 ± 0.26 a | 3.73 ± 0.31 a |
| A6 | 450.0 | 4.53 ± 0.11 a | 3.31 ± 0.22 a |

Different lowercase letters indicate that the difference between treatments is 0.05 significance level. Data is displayed with mean ± SE.

### 3.1.2. Dynamics of the Plant Population

With the increase of planting density, the number of population stems showed an upward trend in each period (Table 2). Under A4, A5 and A6 conditions, the total stem number of hybrid wheat before winter was more than $1000 \times 10^4$ ha$^{-1}$. The population stem number showed a trend of first increasing and then decreasing with the development period, with the highest stem number at the rising stage. Under $225–375 \times 10^4$ ha$^{-1}$ conditions, the total number of stems was higher, which indicated that the high-density population of hybrid wheat was easy to cause degradation of the early tillers. When the seeding density exceeds $225 \times 10^4$ ha$^{-1}$, the total stem number of hybrid wheat at jointing stage exceeded $1500 \times 10^4$ ha$^{-1}$, which indicates that hybrid wheat has outstanding tillering ability, especially under relatively low planting density. As shown in the table, the density is less than $300 \times 10^4$ ha$^{-1}$, the tiller–spike rate was higher than 20%, with an average of 26.0%. If the planting density is higher than $300 \times 10^4$ ha$^{-1}$, the tiller–spike rate was only 12.6%. It can be concluded that under the condition of medium- and lower-density planting, the higher tiller–spike rate is conducive to the construction of a reasonable population of hybrid wheat.

**Table 2.** Dynamics of wheat plant population quantity in different densities ($\times 10^4$ ha$^{-1}$).

| Treatments | Basic Seedling ($\times 10^4$ ha$^{-1}$) | Growth Stage | | | | | The Tiller–spikeRate |
|---|---|---|---|---|---|---|---|
| | | Overwintering Period | Regreening Period | Raising Period | Jointing Period | Maturity Period | |
| A1 | 75.0 | 304.5 ± 12.7 f | 301.5 ± 13.4 d | 1368.8 ± 103.9 c | 862.5 ± 55.3 c | 562.5 ± 22.5 b | 37.7 ± 3.6 a |
| A2 | 150.0 | 592.5 ± 34.6 e | 577.5 ± 25.3 c | 1862.5 ± 99.8 b | 1293.8 ± 90.4 b | 575.0 ± 14.7 b | 24.8 ± 2.7 b |
| A3 | 225.0 | 852.0 ± 25.1 d | 798.0 ± 25.8 b | 2150.0 ± 93.6 a | 1582.5 ± 97.3 a | 637.5 ± 37.7 a | 21.4 ± 2.5 b |
| A4 | 300.0 | 1042.5 ± 76.4 c | 981.0 ± 41.3 b | 2198.8 ± 78.5 a | 1635.0 ± 62.5 a | 682.8 ± 31.0 a | 20.2 ± 2.0 b |
| A5 | 375.0 | 1297.5 ± 119.6 b | 1231.5 ± 51.4 a | 2225.0 ± 85.7 a | 1627.5 ± 76.1 a | 664.6 ± 23.4 a | 15.7 ± 2.1 c |
| A6 | 450.0 | 1434.0 ± 78.5 a | 1342.5 ± 65.9 a | 2000.0 ± 104.6 a | 1642.5 ± 88.7 a | 645.3 ± 27.8 a | 12.6 ± 1.4 c |

Different lowercase letters indicate that the difference between treatments is 0.05 significance level. Data is displayed with mean ± SE. Overwintering period, when the average temperature before winter in the main winter wheat production area of China drops below 0 °C~1 °C, almost all wheat seedlings will stop growing. Regreening period, when the temperature rises in spring, the heart leaves are gradually drawn out. Raising period, the shape of wheat changed from creeping to upright, and the first internode of the stem began to elongate but did not protrude from the ground (Zadoks, GS30). Jointing period, when the first internode of the main stem of more than 50% of wheat plants in the whole field is 1.5 cm~2 cm above the ground (Zadoks, GS31). Maturity period, when the endosperm of wheat is waxy and the grain begins to harden (Zadoks, GS92).

### 3.2. Material Accumulation Characteristics of Hybrid Wheat

Dry matter production, especially the photosynthetic production during the yield formation period, is an important indicator to measure the quality of the population [34–37]. With the increase of sowing density (Table 3), there was no obvious linear relationship between the material accumulation after anthesis and planting density of hybrid wheat. The material accumulation of A4, A5 and A6 treatments was significantly higher than that of A1, A2 and A3 treatments at maturity.

**Table 3.** Dynamics of dry matter accumulation in different density treatments (kg ha$^{-1}$).

| Treatments | Basic Seedling (×10⁴ ha⁻¹) | Overwintering Period | Jointing Period | Booting Period | Anthesis Period | Maturity Period | Dry Matter after Anthesis | Relative Proportion of Dry Matter Pre-Anthesis | Relative Proportion of Dry Matter after Anthesis |
|---|---|---|---|---|---|---|---|---|---|
| A1 | 75.0 | 552.4 ± 23.4 d | 1221.0 ± 76.5 d | 3488.0 ± 124.5 d | 6124.5 ± 223.9 c | 10831.5 ± 477.8 b | 4707.0 ± 204.2 ab | 56.5 ± 0.5 c | 43.5 ± 0.6 a |
| A2 | 150.0 | 724.5 ± 56.9 b | 2313.0 ± 145.3 c | 5008.0 ± 223.1 c | 6937.7 ± 304.2 c | 11481.8 ± 689.1 b | 4544.1 ± 178.9 b | 60.4 ± 1.1 b | 39.6 ± 1.2 b |
| A3 | 225.0 | 732.1 ± 45.6 b | 3880.5 ± 155.8 b | 6288.0 ± 245.3 b | 8374.5 ± 563.2 b | 12844.5 ± 783.9 b | 4470.0 ± 198.4 b | 65.2 ± 0.9 a | 34.8 ± 0.3 c |
| A4 | 300.0 | 679.4 ± 51.1 c | 3964.5 ± 126.7 b | 6656.0 ± 378.5 b | 8991.0 ± 365.4 b | 14154.8 ± 798.3 a | 5163.8 ± 205.3 a | 63.5 ± 1.3 ab | 36.5 ± 0.8 bc |
| A5 | 375.0 | 866.6 ± 76.3 a | 4059.0 ± 98.2 b | 8040.0 ± 357.2 a | 10281.0 ± 643.7 a | 15204.0 ± 899.4 a | 4923.0 ± 177.4 a | 67.6 ± 1.9 a | 32.4 ± 0.3 c |
| A6 | 450.0 | 695.0 ± 45.2 c | 5176.5 ± 195.1 a | 8064.0 ± 403.6 a | 10631.3 ± 732.2 a | 15795.0 ± 766.2 a | 5165.0 ± 161.2 a | 67.3 ± 2.4 a | 32.7 ± 0.5 c |

Different lowercase letters indicate that the difference between treatments is 0.05 significance level. Data is displayed with mean ± SE. Overwintering period, when the average temperature before winter in the main winter wheat production area of China drops below 0 °C~1 °C, almost all wheat seedlings will stop growing. Jointing period, when the first internode of the main stem of more than 50% of wheat plants in the whole field is 1.5 cm~2 cm above the ground (Zadoks, GS31). Booting period, more than half of the flag leaves of wheat ears in the field were formally drawn out from the leaf sheaths, and the parts wrapping the young ears were significantly expanded (Zadoks, GS45). Anthesis period, the inner and outer glumes of the upper and middle florets of more than 50% wheat ears in the whole field opened, and the anthers began to disperse pollen (Zadoks, GS65). Maturity period, when the endosperm of wheat is waxy and the grain begins to harden (Zadoks, GS92).

### 3.3. Dry Matter Productivity

The dry matter stored before anthesis of wheat is very important for grain filling [38–42]. The transport and utilization of pre-anthesis storage materials of hybrid wheat varied with planting density (Table 4). The average transfer rate and contribution rate of pre-anthesis storage materials of hybrid wheat were 12.4% and 17.2%, respectively, which were the highest under A3 condition. The average material productivity of hybrid wheat after anthesis was only 117.6 g/m$^2$, and it decreased with the increase of density, which indicates that different planting density affects the balance of source–sink relations of hybrid wheat, and the function of "source" is relatively insufficient after anthesis.

**Table 4.** The effects of density treatments to dry matter productivity.

| Treatments | Reserve Per-Anthesis | | | Dry Matter Productivity after Anthesis (g/m² leaf) |
|---|---|---|---|---|
| | Transportation Amount (kg ha⁻¹) | Transfer Percentage (%) | Contribution Proportion (%) | |
| A1 | 788.4 ± 36.5 c | 12.9 ± 0.4 b | 14.3 ± 0.4 c | 133.3 ± 5.2 a |
| A2 | 988.3 ± 39.6 b | 14.2 ± 0.8 a | 17.9 ± 1.7 b | 118.3 ± 4.6 b |
| A3 | 1303.2 ± 65.3 a | 15.6 ± 1.0 a | 22.6 ± 1.4 a | 110.1 ± 4.4 b |
| A4 | 973.4 ± 41.2 b | 10.8 ± 0.4 c | 15.9 ± 1.2 b | 120.9 ± 6.5 b |
| A5 | 1155.2 ± 54.9 a | 11.2 ± 0.2 c | 19.0 ± 1.9 b | 103.6 ± 1.7 c |
| A6 | 829.3 ± 29.9 c | 7.8 ± 0.9d | 13.8 ± 0.54 c | 114.0 ± 2.13 b |

Different lowercase letters indicate that the difference between treatments is 0.05 significant level. Data is displayed with mean ± SE.

### 3.4. Yield Composition and Source–Sink Relations of Hybrid Wheat

With the increase of sowing density, the yield of hybrid wheat increased first and then decreased, among which A3 had the highest yield (Table 5). Lower planting density (75 × 10⁴ ha⁻¹) or higher (375 × 10 ha⁻¹) were not conducive to the increase of yield. Under A2 and A3 conditions (Table 6), the spike–leaf area ratio and grain-to-leaf ratio were higher, which indicated that hybrid wheat mainly relied on tillers to obtain high sink–source ratio under low density conditions. The dry matter mass-per-unit grain pre-anthesis increased with the increase of sowing density (Table 6), and the dry matter mass-per-unit grain after anthesis decreased first and then increased. On the whole, the two indicators of A2 and A3 treatment are lower, which may be due to the high total storage capacity and insufficient material accumulation. These further highlight the importance of strengthening the source function, and "increasing source and strengthening source" is the key to further improve the sink (grains) filling.

**Table 5.** Yield components of hybrid wheat in different density treatments.

| Treatments | Basic Seedling ($\times 10^4$ ha$^{-1}$) | Spikes ($\times 10^4$ ha$^{-1}$) | Grain No. per Spike | Grain Weight (mg/Kernel) | Grain Yield (kg ha$^{-1}$) |
|---|---|---|---|---|---|
| A1 | 75.0 | 562.5 ± 20.8 b | 29.5 ± 0.5 a | 38.0 ± 0.2 a | 5495.4 ± 113.7 b |
| A2 | 150.0 | 637.5 ± 13.8 a | 27.3 ± 0.4 b | 40.9 ± 0.1 a | 5773.2 ± 151.2 a |
| A3 | 225.0 | 682.8 ± 14.7 a | 27.7 ± 1.1 b | 40.3 ± 0.4 a | 6137.2 ± 223.5 a |
| A4 | 300.0 | 664.6 ± 15.4 a | 26.2 ± 0.2 b | 39.3 ± 0.5 a | 6078.2 ± 160.6 a |
| A5 | 375.0 | 645.3 ± 19.1 a | 26.1 ± 0.6 b | 40.4 ± 0.3 a | 5993.1 ± 125.7 a |
| A6 | 450.0 | 639.6 ± 17.4 a | 25.9 ± 0.5 b | 40.3 ± 0.2 a | 5797.5 ± 178.3 a |

Different lowercase letters indicate that the difference between treatments is 0.05 significance level. Data is displayed with mean ± SE.

**Table 6.** Characteristics of sink–source system of hybrid wheat with different densities.

| Treatments | Basic Seedling ($\times 10^4$ ha$^{-1}$) | ($\times 10^7$ ha$^{-1}$) Sink Capacity ($\times 10^7$ Grain ha$^{-1}$) | The Spike to Leaf Ratio (Spikes/m$^2$ leaf) | The Grain to Leaf Ratio (Grain/m$^2$ Leaf) | Dry Matter Mass per Unit Grain Pre-Anthesis (mg/Grain) | Dry Matter Mass per Unit Grain after Anthesis (mg/Grain) |
|---|---|---|---|---|---|---|
| A1 | 75.0 | 16.6 ± 1.2 a | 204.3 ± 7.6 b | 6436.6 ± 91.0 b | 36.9 ± 1.9 c | 28.4 ± 0.5 a |
| A2 | 150.0 | 17.4 ± 0.9 a | 228.2 ± 11.3 a | 7721.4 ± 122.5 a | 48.1 ± 2.7 b | 25.7 ± 1.3 b |
| A3 | 225.0 | 18.9 ± 2.2 a | 223.9 ± 9.5 a | 7630.2 ± 104.3 a | 47.6 ± 2.1 b | 27.3 ± 0.5 b |
| A4 | 300.0 | 17.4 ± 1.1 a | 178.0 ± 7.1 c | 6079.5 ± 86.2 b | 59.0 ± 2.3 a | 28.3 ± 0.3 a |
| A5 | 375.0 | 16.9 ± 1.5 a | 194.7 ± 6.0 b | 6282.6 ± 79.3 b | 63.0 ± 3.0 a | 30.6 ± 2.1 a |
| A6 | 450.0 | 16.6 ± 1.3 a | 194.4 ± 5.2 b | 6134.6 ± 106.9 b | 63.0 ± 2.1 a | 30.1 ± 1.7 a |

Different lowercase letters indicate that the difference between treatments is 0.05 significance level. Data is displayed with mean ± SE.

*3.5. The Relations of Yield Components to Source–Sink Indicator*

The coordination of yield components is the basis for obtaining high yield. The correlation analysis showed (Table 7) that the yield of hybrid wheat was positively correlated with the sink capacity ($r$ = 0.764 **), indicating that the sink capacity was the main contributing factor to the yield formation of hybrid wheat. Further analysis of yield components showed that the yield of hybrid wheat was positively correlated with the number of spikes ($r$ = 0.898 **), but not significantly correlated with the number of grains-per-spike and grain weight ($r$ = 0.056, 0.186), which further explained the important role of the number of spikes in stabilizing and improving the yield of hybrid wheat.

**Table 7.** The correlation analysis of yield components and yield of hybrid wheat.

| | Grain Yield | Spikes | Grain No. per Spike | Grain Weight | Sink Capacity | The Spike to Leaf Ratio | The Grain to Leaf Ratio |
|---|---|---|---|---|---|---|---|
| Grain yield | 1.000 | | | | | | |
| Spikes | 0.898 ** | 1.000 | | | | | |
| Grain no. per spike | 0.056 | −0.727 | 1.000 | | | | |
| Grain weight | 0.186 | 0.117 | −0.158 | 1.000 | | | |
| Sink capacity | 0.764 ** | 0.736 ** | 0.071 | 0.059 | 1.000 | | |
| The spike to leaf ratio | 0.193 | 0.168 | 0.547 | 0.483 | 0.320 | 1.000 | |
| The grain to leaf ratio | 0.017 | 0.005 | 0.463 * | 0.559 | 0.474 | 0.952 ** | 1.000 |

Correlation coefficients ($r$) are calculated and ** represent significance at the 0.01 probability level and * represent significance at the 0.05 probability level.

Although the contribution of sink capacity to yield formation has reached a very significant level, the contribution of grain weight is not significant, and the correlation coefficient is small. As an element of yield formation, there is a lack of coordination and balance between hybrid wheat storage capacity and grain weight. On the basis of a certain storage capacity, mining the contribution of grain weight to yield formation is the key to obtain a high yield of hybrid wheat.

## 4. Discussion

　　Reasonable source–sink relations is the basis and guarantee of wheat yield formation [6,17]. By constructing reasonable source--sink relations in field production, the population material production, grain material accumulation and distribution can be promoted [42,43]. As an important way to improve food crops, hybrid wheat has been preliminarily applied in production [44]. At present, the plant height of hybrid wheat varieties is 5–10 cm higher than that of conventional varieties, and the leaf shape of the plants is scattered and wider. In addition, the larger population is easy to cause lodging [45], which is not conducive to the presentation of heterosis. Therefore, it is of great significance to construct a reasonable population to improve the yield of hybrid wheat. The results showed that the material accumulation of hybrid wheat increased with the increase of sowing density. However, even when the LAI of population at anthesis reached the highest, the material accumulation of the high-density population between the anthesis and maturity stage did not show a significant upward trend. It can be inferred that the production capacity of the source is insufficient to a large extent, and the high density and large population structure did not match the high population quality. This experimental study shows that under the lower density condition (A2, A3), the yield, spike-to-leaf ratio and grain-to-leaf ratio of hybrid wheat reach the maximum. It can be inferred that the population structure can be optimized by appropriately reducing the planting density, thus improving the population quality of hybrid wheat and achieving the goal of "increasing and strengthening the source". The research shows that the main driving force for achieving super high yield of hybrid rice is large sink capacity and high biological yield [28]. Our results also showed that the yield of hybrid wheat was positively correlated with sink capacity and spike number, and the increase of spike number could promote the expansion of population capacity. Depending on the superiority of tiller and spike, the contribution of spike number to yield formation has reached a very significant level, which was consistent with the study on common wheat cultivars [46]. Under different density, hybrid wheat could obtain higher storage capacity, which indicates that hybrid wheat shows the characteristics of large and stable storage capacity. In the future, we can make full use of this dominant structure to build a high-quality population structure and further improve the yield level of hybrid wheat. It is very interesting that the material production capacity and dry matter quality per unit grain of hybrid wheat pre-anthesis are lower (14.3–22.6%) when compared with that of conventional winter wheat (24.5–29.1%) [47], and the contribution rate of storage material of pre-anthesis vegetative organs to yield formation is more than 20% under A3, in contrast to other planting density. To a certain extent, it compensates for the effect of insufficient source on yield formation during grain filling stage. In the future, we should also study the regulation of other cultivation factors to promote the pre-anthesis material transporting [48]. The proportion of material accumulation after anthesis showed a downward trend while the total amount of material accumulation was increased by increasing the planting density. We can conclude that the larger population would lead to the relative deficiency of "source" function and insufficient material transporting during the yield formation period. It is also very important for hybrid wheat to control sowing density, mold reasonable population and improve population quality due to the higher tillering capacity of Jingmai 6 under lower plant densities. Our previous studies also showed that hybrid wheat has a high biomass, which indicates that it may have a high stress tolerance [5]. The new hybrid variety JM-1683 showed a yield increase of more than 10% compared with the local varieties in Pakistan under the condition of controlled nitrogen application (the data from the University of Agriculture Peshawar during winter season 2020–2021); however, the high-yield population construction of different types of hybrid wheat under adverse conditions needs further research. Through the preliminary study of this experiment, the Jingmai 6 showed the advantages of more tillers and ears and greater contribution potential of nutrient organ storage material redistribution under lower planting density conditions. However, at present, the study on the differences between hybrid wheat and conventional wheat is only limited to their phenotypic differences [45]. In the future, we should carry out in-

depth research on the molecular physiological mechanism of the differences in source–sink relations and material distribution. From the aspect of variety breeding, the breeding of hybrid wheat should pay more attention to the construction of ideal plant types, so as to maximize the performance of heterosis such as yield and stress resistance [49,50]. From the perspective of high-efficiency cultivation, controlling sowing density for hybrid wheat is an important way to give full play to individual advantages and then increase its yield.

## 5. Conclusions

The characteristics of its source–sink relations and yield composition of Jingmai 6 under different planting densities were studied. This experiment proved that under the condition of medium- and low-planting density ($225$–$300 \times 10^4$ ha$^{-1}$), hybrid wheat showed high-quality population characteristics and reasonable source–sink relations, such as suitable LAI, high tiller–spike rate, higher grain to leaf ratio, etc., which are the key to obtaining a high yield of hybrid wheat under this condition. These results of this experiment also provide data support for the application of other hybrid wheat varieties with the same characteristics.

**Author Contributions:** Z.C. and S.Z. conceived the main idea of the research. W.Y. analyzed the data and wrote the manuscript. Z.W., L.R., Z.Y., X.G., J.G., H.L. and B.D. revised the manuscript and provided suggestions. In addition, Z.W., L.R., Z.Y. and X.G. assessed and data collection. All authors have read and agreed to the published version of the manuscript.

**Funding:** This research was funded by the Science and Technology Innovation Project of BAAFS (KJCX20151403) and China's National Key R&D Programmes (2016YFD0101604).

**Data Availability Statement:** Not applicable.

**Conflicts of Interest:** The authors declare no conflict of interest.

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
