# Peer review of "Effects of Population Regulation on the Source–Sink System of Hybrid Wheat Jingmai 6"

_agronomy, doi:10.3390/agronomy12102530_

Round 1

Reviewer 1 Report

Dear Authors, 

The manuscript is of a technical nature, it does not add any scientific aspect. The reader has the impression that this is the characteristic of one hybrid variety of Jingmai 6 wheat together with the recommendation of the most effective sowing, taking into account the selected growth and yield parameters. This work is important in the conditions of local agriculture, the results and conclusions are not comparable with the results obtained for other hybrid varieties or traditional lines. 

Author Response

Response to Reviewer 1 Comments

Point 1:The manuscript is of a technical nature, it does not add any scientific aspect. The reader has the impression that this is the characteristic of one hybrid variety of Jingmai 6 wheat together with the recommendation of the most effective sowing, taking into account the selected growth and yield parameters. This work is important in the conditions of local agriculture, the results and conclusions are not comparable with the results obtained for other hybrid varieties or traditional lines. 

Response 1: The application of hybrid wheat is a worldwide scientific problem. In 2005, Jingmai 6 was approved by Beijing Variety Approval Committee, and it is the first hybrid wheat variety (two-line method) bred by us. At the time of this study, no more hybrid wheat varieties were approved. Therefore, only one hybrid wheat variety was selected for relevant research. With the increase of the number of approved hybrid wheat, we will increase the comparison between different varieties in the future research in order to obtain more scientific data.

Point 2:The following problems mentioned by experts.

Response 2: Corresponding modifications have been made in the paper (in red)

Reviewer 2 Report

Research results interesting for science and agricultural practice.

Their big drawback is that they come from a one-year experiment.

Such results cannot be the basis for general conclusions about the plant response to changing environmental conditions (weather, soil).

Such results should be published as short communication, not an article.

Comments, questions and suggestions for the submitted article are included in the manuscript.

Author Response

Response to Reviewer 2 Comments

Point 1:Their big drawback is that they come from a one-year experiment. Such results cannot be the basis for general conclusions about the plant response to changing environmental conditions (weather, soil). Such results should be published as short communication, not an article.

Response 1: In 2005, Jingmai 6 was approved by Beijing Variety Approval Committee, and it is the first hybrid wheat variety (two-line method) bred by us. After many years of multi-point tests, the variety approval committee suggested that the suitable basic seedlings should be 100000~150000/667m2. In order to explore the effect of further reducing or increasing planting density on the yield and source-sink system of Jingmai 6, this experiment has made a preliminary exploration. Although the experiment has only been carried out for one year, the test results are  highly consistent with the planting density recommended by the variety approval committee, which proves the reliability of this test results. In addition, this experiment showed the characteristics of more repetitions (4 repetitions) and large planting area (7.2 square meters for each treatment), which to some extent can make up for the shortcomings of less experimental years. These can better ensure the reliability of the test results. Of course, we very much agree with the opinions of the reviewers. In the future trials, with the increase of the number of approved hybrid wheat varieties, we will design multi point comparative trials for different types of varieties for many years to reveal relevant conclusions more scientifically.

Point 2:The problems mentioned in the manuscript by experts.

Response 1: The problems raised by the reviewers in the manuscript have been revised accordingly(in red).

Reviewer 3 Report

Manuscript title:   Effects of population regulation on the source-sink system of  hybrid wheat

Authors:  Weibing Yang, et al.

The manuscript regarding the topic and results presented is of interest to the hybrid wheat scientific community and revisions based on the comments below are recommended before considering for publication.

Abstract: In the abstract, the main aim and background of the manuscript is missing. In addition, it would be even better to have a sentence as a future perspective.

Introduction: I suggest to the authors supplement the Introduction with the current field research about the yielding of hybrid wheat, conducted not only in China but for example, in Europe or in the USA, especially since rightly according to the authors’ claim hybrid wheat is considered to be one of the main ways to greatly improve the wheat yield in the future.

Line 61-65: That seems a poorly structured hypothesis, please recheck.

Materials and Methods

Plant materials and experimental design: Field experiment results should be at least 2 or 3 years, not just 1 year ?? unless I misunderstand the record.

Line 76-80: It would be better to change the text to a scheme or diagram to illustrate the field experiments

Measurements and Analyses: Literature references are missing for all sub-section  2.2.1.; 2.2.2.; 2.2.3. and 2.2.4. It would be better to cite the references that the procedure adopted.

Line 92-93: Why, in the case of LAI measurements, did not use generally available stationary or field measuring instruments?

Results: I suggest in the Results chapter also insert the cited table with these results and delete chapter 3.6., which only contains tables.

Table 2 and Table 3, the names of the growth stages of wheat should be given once more, the current names of the development stages are incomprehensible.

Discussion: The Discussion (especially from Line 253 to Line 278) should be supplemented by quoting new literature items.

Conclusion: I believe there are other a lot nice conclusions could be made from this manuscript…What are the prospects for hybrid wheat cultivation in China?

Author Response

Response to Reviewer 3 Comments

Point 1:Abstract: In the abstract, the main aim and background of the manuscript is missing. In addition, it would be even better to have a sentence as a future perspective.

Response 1: The sentence of “Hybrid wheat is considered to be one of the main ways to greatly improve the wheat yield in the future” were added and have been revised accordingly.  (In red)

Point 2:Introduction: I suggest to the authors supplement the Introduction with the current field research about the yielding of hybrid wheat, conducted not only in China but for example, in Europe or in the USA, especially since rightly according to the authors’ claim hybrid wheat is considered to be one of the main ways to greatly improve the wheat yield in the future.

Response 2: The current field research about the yielding of hybrid wheat in Europe was added to the part of “introduction” .

Point 3:Line 61-65: That seems a poorly structured hypothesis, please recheck.

Response 3: Revised accordingly (In red)

Point 4:Materials and Methods. Plant materials and experimental design: Field experiment results should be at least 2 or 3 years, not just 1 year ?? unless I misunderstand the record.

Response 4: Jingmai 6 was approved by Beijing Variety Approval Committee, and it is the first hybrid wheat variety (two-line method) bred by us. After many years of multi-point tests, the variety approval committee suggested that the suitable basic seedlings should be 100000~150000/667m2. In order to explore the effect of further reducing or increasing planting density on the yield and source-sink system of Jingmai 6, this experiment has made a preliminary exploration. Although the experiment has only been carried out for one year, the test results are  highly consistent with the planting density recommended by the variety approval committee, which proves the reliability of this test results. In addition, this experiment showed the characteristics of more repetitions (4 repetitions) and large planting area (7.2 square meters for each treatment), which to some extent can make up for the shortcomings of less experimental years. These can better ensure the reliability of the test results. Of course, we very much agree with the opinions of the reviewers. In the future trials, with the increase of the number of approved hybrid wheat varieties, we will design multi point comparative trials for different types of varieties for many years to reveal relevant conclusions more scientifically.

Point 5:Measurements and Analyses: Literature references are missing for all sub-section  2.2.1.; 2.2.2.; 2.2.3. and 2.2.4. It would be better to cite the references that the procedure adopted.

Response 5: Literature references are added accordingly.

Point 6:Line 92-93: Why, in the case of LAI measurements, did not use generally available stationary or field measuring instruments?

Response6: The title of our references is” Canopy morphological changes and water use efficiency in winter wheat under different irrigation treatments”

Point 7:Results: I suggest in the Results chapter also insert the cited table with these results and delete chapter 3.6., which only contains tables.

Response 7: This part is arranged according to the requirements of this publication. If the paper is accepted, the journal editor will revise it according to the experts' opinions.

Point 8:Table 2 and Table 3, the names of the growth stages of wheat should be given once more, the current names of the development stages are incomprehensible.

Response 8: added accordingly.

Point 9:Discussion: The Discussion (especially from Line 253 to Line 278) should be supplemented by quoting new literature items.

Response 9: added accordingly.

Point 10:Conclusion: I believe there are other a lot nice conclusions could be made from this manuscript…What are the prospects for hybrid wheat cultivation in China?

Response 10: added accordingly in the part of discussion .

Round 2

Reviewer 1 Report

Dear Authors,

I still think that the manuscript has the character of technical information, more of an information pamphlet than a scientific work. In addition, in scientific work, the factor of reproducibility of the results obtained is important, I am afraid that after one year, the results of a field experiment may not be reproducible. At least 2 or 3 years are needed, especially in a new variety line.

If this work can be published, a comment should be added in the title that this is information about the first hybrid wheat line in China. However, in this case, the work is local, so I suggest including more data about hybrid wheat lines in the world in the content and/or changing the manuscript type. I also suggest, despite only results from one year, to relate these results to other parameters obtained from an another hybrid wheat line grown in the region of the world with, for example, similar climatic conditions. 

Author Response

Thank you very much for your suggestions. First of all, we revised the title of the paper. Moreover, we added the yield increase of our latest hybrid wheat combination (JM-1683)under nitrogen control conditions in Pakistan to the discussion section. The above revisions provide basic data for understanding hybrid wheat.

Reviewer 3 Report

Dear Authors

Due to the research on a new hybrid variety of wheat, I accept exceptionally 1-year field research.

Best regards

Author Response

Thank you for your suggestion